# Comparison of glenohumeral joint kinematics between swimmers clinically classified with multidirectional instability and asymptomatic controls

Oliver A. Silverson[ID][○], Gaura Saini[ID][○], Ward M. Glasoe[○], Paula M. Ludewig[○], Justin L. Staker[○]*

Minnesota Rehabilitation Biomechanics Laboratory, Division of Physical Therapy and Rehabilitation Science, University of Minnesota Medical School, Minneapolis, Minnesota, United States of America

○ These authors contributed equally to this work.
* stak@umn.edu

## Abstract

The clinical classification of glenohumeral joint instability is characterized by presumed increased humeral translations in conjunction with symptoms of instability. Prior research reports inconsistent kinematic differences in glenohumeral kinematics between individuals clinically classified with multidirectional instability and asymptomatic controls. Differing clinical classifications and motion tracking methods likely contribute to this gap. This analysis aimed to compare three-dimensional (3D) glenohumeral joint kinematics during active arm raising between individuals clinically classified with multidirectional instability and asymptomatic matched controls. Twenty competitive swimmers (13 female; mean age: 24.85; standard deviation (SD): 12.51) clinically classified with multidirectional instability via a comprehensive clinical examination and 10 asymptomatic matched controls (6 female: mean age: 24.70; SD: 7.04) were enrolled. Active, unweighted, scapular plane abduction was recorded with dynamic biplane video radiography, and glenohumeral joint kinematics were reconstructed with 2D/3D shape-matching. The variables compared between groups included: humeral position along the anterior/posterior and superior/inferior axes of the glenoid, positional dispersion of the humeral instantaneous helical axis, and humeral contact path length on the glenoid. The average humeral position between 30°-90° of glenohumeral elevation was significantly more anterior (+0.8 mm, $P < 0.001$, effect size = 0.57) in individuals classified with multidirectional instability compared to controls. No other significant differences were detected. Our findings indicate that individuals classified with multidirectional instability possess significantly greater average humeral head position in the anterior direction. However, these individuals do not possess markedly different glenohumeral joint kinematics in superior/inferior humeral position, humeral instantaneous helical axis positional dispersion, or humeral contact path length compared to asymptomatic individuals

**Data availability statement:** All relevant data are within the paper and its Supporting Information files.

**Funding:** Funding Funding mechanism followed by author initials or institution responsible for acquiring funding: Grand-In-Aid Award, American Society of Biomechanics (OS). https://asbweb.org/ Grant-in-Aid of Research, Artistry and Scholarship Grant (#467743) from the Office of the Vice President for Research, University of Minnesota (JS). https://research. umn.edu/ Promotion of Doctoral Studies II Scholarship, Foundation for Physical Therapy Research (GS). https://foundation4pt.org/ portfolio_category/pods-ii/ National Institute of Health/National Institute of Arthritis and Musculoskeletal and Skin Diseases, F31 Ruth L. Kirschstein Predoctoral Individual National Research Service Award (#1F31AR079259) (GS). https://researchtraining.nih.gov/pro-grams/fellowships/F31 Midwest Center for Occupational Health and Safety - Education and Resource Center, Pilot Projects Research Training Program (GS). https://mcohs.umn. edu/ National Institutes of Health's National Center for Advancing Translational Sciences, grant UL1TR002494 (University of Minnesota). The content is solely the responsibility of the authors and does not necessarily represent the official views of the National Institutes of Health's National Center for Advancing Translational Sciences. https://ncats.nih.gov/ funding/funding-opportunities The funders had no role in study design, data collection and analysis, decision to publish, or preparation of the manuscript.

**Competing interests:** The authors have declared that no competing interests exist.

during unweighted arm elevation. Further exploration is necessary to identify novel kinematic variables that accurately quantify group differences in joint stability.

## Introduction

The presumed presence of increased humeral head translations in conjunction with patient-reported symptoms characterizes the clinical construct of glenohumeral joint instability [1,2]. Symptoms of glenohumeral joint instability can range from upper extremity pain during activity or rest to mistrust or feelings of apprehension during activities [3]. Assessment often combines clinical glenohumeral joint laxity testing and evaluation of patient-reported symptoms [4,5]. The traditional classifications of glenohumeral joint instability describe instability in two or more directions as multidirectional instability (MDI) [2]. Accordingly, it is often assumed that "stable" shoulders possess minimal humeral translations while shoulders with MDI, or "unstable" shoulders, possess increased humeral translations during dynamic tasks [6–8], however, this premise requires examination.

Prior studies have attempted to identify distinct dynamic kinematic patterns in individuals with MDI compared to asymptomatic individuals, yet findings have been inconsistent. Illyés and Kiss [9] reported that individuals clinically classified with MDI possessed significantly greater average anterior (0.07 mm more) and superior (0.13 mm more) displacement between the humeral and scapular centers of rotation during scapular plane abduction (SAB) compared to asymptomatic control individuals. However, the clinical implications of differences less than 1.0 mm remain unclear. In contrast, Ogston and Ludewig [10] did not detect statistically significant differences in humeral translation, measured via helical translations, across 30° intervals of SAB between individuals clinically classified with MDI and matched controls. Other research [11–13] also reported inconsistent findings concerning the magnitude and direction of three-dimensional (3D) glenohumeral joint kinematics between groups, with reported differences in average 3D humeral translations ranging up to 3.0 mm in conflicting directions. These discrepancies likely stem from differences in the clinical classification of glenohumeral joint instability, participant exposure to shoulder activity, limitations surrounding surface-based motion tracking or static imaging techniques, and kinematic description approaches.

Implementing standardized criteria to evaluate glenohumeral joint laxity with associated symptomatology may more accurately classify individuals with MDI. Additionally, analysis of kinematics collected from individuals with various exposure to shoulder activity in recreation or occupation may uncontrollably increase the variability in intrinsic kinematic patterns of the sample, thereby making a consistent pattern difficult to detect. To overcome this limitation, potentially capturing kinematics from participants with similar exposures to overhead activities, such as competitive swimmers, may improve the ability to distinguish the kinematic patterns suspected of MDI. Competitive swimmers have been demonstrated to possess consistent training habits and clinical movement characteristics [14–16]. Related to the motion capture method, skin motion artifacts may affect kinematics collected with surface-based motion

recording systems [17] and cannot depict joint surface-level interactions. More precise motion capture methods could overcome these limitations, such as dynamic biplane radiography combined with 2D/3D shape-matching, with reported tracking errors between 0.5–0.9 mm and 0.5–2.0° for the humerus and scapula, respectively (20). Lastly, quantifying glenohumeral joint kinematics with 2D, planar humeral translations may only partially account for the subtle movement patterns suspected to occur in three dimensions simultaneously in people with glenohumeral joint instability. Alternative kinematic description techniques, such as helical axis parameters and arthrokinematic variables of contact patterns, may allow a more comprehensive analysis of joint stability [18]. A helical axis is a method to capture a rigid body's rotation about and translation along a line in 3D space [19]. In contrast, a contact pattern defines the surface-to-surface relationship between two surfaces [20]. The humeral helical axis positional dispersion, or distribution of helical axes relative to a fixed point, may offer insight into humeral stability across motion.

Our objective was to compare the 3D glenohumeral joint kinematics during active SAB between competitive swimmers clinically classified with MDI and matched controls. We used dynamic biplane radiography and 2D/3D registration methods to capture and reconstruct 3D kinematics accurately. We hypothesized that individuals classified with MDI would demonstrate significantly different kinematic patterns compared to age and sex matched controls.

## Methods

This was a cross-sectional study performed in a laboratory setting, approved by the University of Minnesota Institutional Review Board (IRB#: STUDY00008772 & 1603M85761).

### Participants

Participants were recruited through outreach to local swimming communities using digital and physical announcements, including targeted email campaigns, community flyers, and social media communications. Recruitment started on 01 September 2021 and ended on 28 February 2023. All participants were enrolled and completed study activities related to this analysis within this timeframe. The study enrolled two distinct cohorts: a clinical instability group composed of competitive swimmers diagnosed with multidirectional instability (MDI), and a control group of asymptomatic individuals without clinical signs of glenohumeral joint laxity. Post-hoc group assignment for matched controls was based on alignment with the clinical group regarding age, height, weight, and body mass index.

An *a priori* power analysis indicated that 10 participants per group would be needed to detect a clinically significant difference in glenohumeral kinematics (minimal clinically important difference (MCID) of 1.0 mm humeral translation with an expected variation ± 0.5 mm) with 80% power. The MCID used in the power analysis was determined from previously reported data comparing 3D glenohumeral joint positions between individuals with shoulder pain and asymptomatic controls [21]. The power calculation was conducted in RStudio Version 3.6.2 (The R Foundation for Statistical Computing, Vienna, Austria), using the "pwr" package (version 1.3, Champely, 2020).

### Clinical instability group

Participants for the clinical instability group were enrolled in a larger, ongoing research program investigating shoulder biomechanics and instability in competitive swimmers [22] which included cross-sectional and interventional analyses. No intervention was completed prior to the current study. All activities related to the current analysis by the clinical instability group were completed during the recruitment period stated previously. Inclusion criteria for the clinical instability group required participants to swim train ≥6 hours/week, be currently or formerly coached for ≥3 years, and compete in ≥1 sanctioned swim event in the past 12 months. Individuals meeting these criteria were screened for the presence of MDI based on the following: an individual was clinically classified as having MDI if they were determined to have excessive multidirectional glenohumeral joint laxity and reported symptoms associated with glenohumeral joint instability in the same shoulder. A battery of standardized clinical tests was administered to evaluate glenohumeral

joint laxity, including the anterior and posterior drawer tests, the sulcus sign, the apprehension test, and the Beighton score for generalized joint hypermobility. The drawer and sulcus tests were each scored according to established grading criteria [23,24], and their average was used to generate a composite laxity index ranging from 0 to 3 [25]. This approach has demonstrated strong inter-rater reliability (ICC > 0.84) [25]. The apprehension test and Beighton Index were assessed following standard protocols [26,27]. A shoulder was categorized as having excessive multidirectional laxity if the composite score met or exceeded 1.6 and the individual presented with either a positive apprehension response or a Beighton score of at least 2 [28]. Preliminary evidence suggests that this classification framework can differentiate between individuals with instability and asymptomatic controls, as demonstrated through dynamic imaging of passive joint translations during manual clinical laxity tests [29]. Once identified with increased laxity, participants were prospectively observed across their respective training seasons for signs and symptoms consistent with multidirectional instability.

Symptoms associated with MDI were evaluated via symptom surveys, which were emailed to participants classified with increased laxity each week once enrolled. Each survey captured the number of days participants had shoulder pain, a visual analog scale, and the Western Ontario Shoulder Instability (WOSI) questionnaire [30]. The study team monitored the results of surveys upon completion. A participant was categorized with symptoms of instability if they met at least one of the following criteria: shoulder discomfort lasting ≥6 consecutive days, a ≥ 20% increase in VAS compared to the previous week, or a ≥ 10% decrease in WOSI score compared to the prior week. These cutoffs were selected under the assumption that shoulder pain lasting over 5 days exceeds the commonly accepted duration of muscle soreness [31], > 20% VAS increase exceeds a clinically meaningful change for shoulder pain [32] and a 10% change in the WOSI is beyond the standard error of measure (8.3%) [33]. Kinematic data were captured during the same week as symptoms beyond the thresholds were first reported. The clinical laxity tests were repeated before motion capture. The shoulder classified with MDI was imaged during motion capture.

### Matched control group

Inclusion criteria for the control group required participants to report no shoulder symptoms in the past year and not have a history of neck or shoulder fractures or surgeries. Additionally, participants were excluded from the control group if they possessed ≤120° of humerothoracic elevation, a composite glenohumeral joint laxity score ≥1.6, or demonstrated abnormal upper extremity movement patterns, such as scapular winging during repeated arm elevation [34]. Competitive swimmers without increased glenohumeral joint laxity and no shoulder pain were allowed to participate in the control group. The shoulder imaged for the control group was selected by the participant.

For both groups, individuals were ≥18 years old and were excluded if they were pregnant or breastfeeding due to exposure to ionizing radiation during motion capture. Study personnel explained all procedures and rationale to participants before enrollment. All enrolled participants provided written consent.

### Data collection

#### Participants

The study team collected each participant's basic demographic information and passive measurements of humerothoracic internal (IR) and external rotation (ER) range of motion (RoM). They recorded measurements using a standard goniometer, with participants positioned supine, the arm abducted to 90°, and the scapula stabilized [35]. Previous research [36] has demonstrated that this technique carries high intra-rater reliability (ICC: > 0.91). These goniometric measures were included to provide clinically relevant descriptors of shoulder movement, allowing for clearer characterization of study groups. This study's glenohumeral analyses (primary analyses) included data only from the shoulder captured during imaging.

## Dynamic motion capture

In vivo glenohumeral joint motion was captured at 60 Hz using a dual-plane video radiography imaging system (Imaging Systems and Services Inc, St. Painesville, OH, USA) during the elevation phase of active, unresisted scapular plane abduction (SAB) (Fig 1). Scapular plane abduction was selected as the test motion to align with existing literature and provide a consistent basis for evaluating the novel kinematic variables introduced in this study. Each trial began with the participant sitting on a stool, feet on the floor, and arms at their sides in a neutral position. Participants raised the imaged arm to maximal elevation in the scapular plane. The research team explained and demonstrated all components, and participants were allowed to practice before motion capture. For the clinical instability group, the arm imaged was classified with MDI. For the healthy cohort, the imaged arm was based on participant preference.

Radiographic motion imaging was performed with tube settings ranging from 70–85 kV and 100–125 mA, using a brief exposure duration of 3.57 ms. The mean effective dose across participants ranged from 0.16 to 1.89 mSv. Calibration and image preprocessing were performed in XMALab (Brown University, Providence, RI, USA: https://www.xromm.org/xmalab/) [37].

## 3D image analysis

Three-dimensional participant-specific bone models of the humerus and scapula were manually reconstructed from computed tomography (CT) (clinical instability group) or magnetic resonance (MR) (control group) images with Mimics image processing software (Mimics, Materialise NV, Leuven, Belgium). CT imaging was obtained for the clinical instability group as part of a broader research program investigating glenohumeral joint instability, described in a previously implemented project from our group [22]. MR imaging was obtained on the control group based on IRB approval for non-interventional research activities. CT scans were acquired with participants' arms at their side in neutral rotation using a Siemens Biograph micro-CT 64-slice flow motion scanner (Siemens Healthineers, Erlangen, Germany) with a 0.6 mm

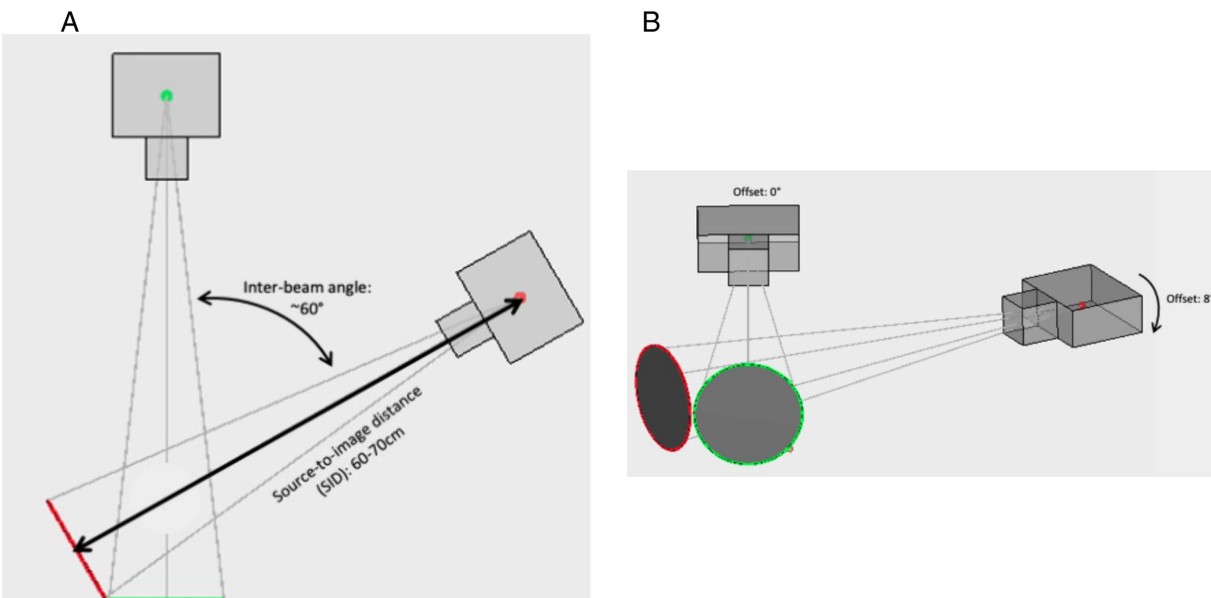

**Fig 1. Dynamic Biplane Video Radiography Orientation.** Caption: 1-A: Top view: the source-to-image distance was 60-70 cm, and the inter-beam angle was approximately 60°. 1-B: Side view: One radiography unit was parallel to the floor while the other was angled upwards 8° to increase the scapular volume in the field of view.

slice thickness (45 kV, 120 mA) and processed with a Siemens B30-Smooth Kernel Filter. MR images were obtained in the same position with a Siemens MAGNETOM Prisma 3 Tesla MRI scanner (Siemens Healthineers, Erlangen, Germany) with an ultra-short echo time 3D gradient with a 0.8 mm slice thickness. Clinical instability group video radiography imaging for this study occurred within a data collection session for another study [22], however, the data reported presently are distinct from those presented elsewhere.

## Data processing

### 2D/3D shape-matching

The 3D positions for the humerus and scapula were obtained via the 2D/3D semi-automated shape-matching process in Autoscoper (Autoscoper, Brown University, Providence, RI, USA: https://simtk.org/projects/autoscoper#) [38] by the primary investigator (OS). Briefly, each bone model was manually manipulated over the 2D radiographic images until the bone edges were aligned in both views. Custom Sobel and contrast filters were applied to radiographic data to enhance the capability of a best-fit solution [38]. A 3-frame moving average filter was applied to the tracked image data. Previous data from our laboratory indicate this approach has glenohumeral joint tracking errors of <0.9 mm and <2.0° [39]. With pilot data, excellent intra-rater reliability: glenohumeral position ($ICC_{3,1} > 0.91$, standard error of measurement (SEM): 0.27–0.52 mm) and glenohumeral rotation ($ICC_{3,1} > 0.94$, SEM: 0.40–2.26°), was established for the primary investigator.

### Kinematic data processing

Kinematic data were processed using a custom biomechanical analysis suite (KinematicsToolbox v4.6, R. Lawrence) [40]. A glenoid-based coordinate system was defined at the glenoid center using rim landmarks following the method of Bey et al. [18]. For participants in the clinical instability group, full humeral reconstructions from CT allowed coordinate system generation via standard anatomical landmarks recommended by the International Society of Biomechanics (ISB) [41]. In contrast, the control group's humeral reconstructions were limited to the proximal third. Thus, a geometric regression model, incorporating participant height and local bone morphology [42], was used to approximate the epicondylar midpoint. A fixed retroversion correction of 57° was then applied to align coordinate axes with expected humeral torsion [43]. Humeral motion was resolved relative to the glenoid using an XZ'Y" Euler angle sequence [44].

Before analysis, joint angle and positional signals were low-pass filtered using a 4th-order Butterworth filter (cutoff: 5 Hz). Each motion trial was downsampled to glenohumeral elevation bins at 5° increments (±2° tolerance) to standardize phase-wise comparisons. All outputs were referenced to a right-handed coordinate system consistent with ISB conventions [41].

In this analysis, we also quantified two joint-level metrics designed to capture subtle features of joint stability: the humeral instantaneous helical axis (IHA) and the contact path length (CPL) between articular surfaces, as previously implemented in a related investigation of treatment response in individuals with MDI (Silverson et al. manuscript in review). Humeral IHA was calculated from the change in points of minimum velocity between radiographic frames [45]. IHA positional dispersion, represented by the magnitude of change in position (mm) from the point closest to all IHAs captured across an interval of motion, was calculated as the root-mean-square of the average axis variability from the optimal pivot location and used for comparison between groups [19]. Humeral CPL was calculated as the overall distance (mm) that the contact center between regions of interest (humeral head and glenoid face) traveled across motion [18]. The contact center was calculated from the weighted average of Euclidean minimum distances between surfaces [20]. To account for inter-individual variation in glenoid anatomy, both measures were scaled to the vertical dimension of the glenoid.

## Statistical analysis

### Participants

Descriptive statistics summarized demographic and clinical movement characteristics. Two-sample *t*-tests were used to compare continuous demographic variables between groups.

## Glenohumeral joint kinematics

The four dependent variables were humeral position relative to the glenoid center along the (1) anterior/posterior and (2) superior/inferior axes, (3) humeral helical axis positional dispersion, and (4) CPL of the humerus traveling on the glenoid. Humeral positions were extracted at 30°/45°/60°/75°/90° of glenohumeral elevation. Data below 30° of glenohumeral elevation was not analyzed as not all participants started in a true 0° position due to the trunk position. Not all participants achieved maximal elevation due to symptoms. Without thoracic kinematics, a 2:1 ratio of humerothoracic to glenohumeral elevation was assumed, in which 90° of glenohumeral elevation equates to roughly 135° humerothoracic elevation [46]. The positional dispersion and CPL magnitude were extracted from each participant's first 50% and second 50% intervals of elevation.

Humeral positions on the anterior/posterior and superior/inferior axes were independently compared between groups with type III two-factor ANOVAs with factors of the group (clinical instability/control) and glenohumeral elevation angle (30°/45°/60°/75°/90°). Multiple imputation via linear predicted values was implemented using the "mice" package (version 3.15, van Buuren, 2022) in RStudio to account for angles with missing data (≤25% of all data) [47]. Positional dispersion and CPL were independently compared between groups with type III two-factor ANOVA with factors of the group (clinical instability/control) and phase of elevation (first 50%/second 50%).

All data were determined to be normal and equally distributed. In the presence of a significant interaction, a Tukey post hoc pairwise test comparing groups at each angle or phase, respectively, was conducted. For all comparisons, in the absence of a significant interaction, the main effect of group was independently evaluated. As an exploratory analysis, the main effect of the elevation phase was also explored using positional dispersion and CPL. Cohen's effect size ($d$) was determined for all main effects and interpreted as $d = 0.2$ is a small effect, $d = 0.5$ is a medium effect, and $d = 0.8$ is a large effect [48].

Statistical analyses were conducted in RStudio Version 3.6.2 (The R Foundation for Statistical Computing, Vienna, Austria), with alpha set a priori at 0.05.

# Results

## Participants

Twenty competitive swimmers were enrolled in the clinical instability group, and ten asymptomatic participants were in the control group. Two participants in the control group were competitive swimmers who had no symptoms or increased glenohumeral joint laxity. There were no significant differences in demographic variables between the groups (Table 1). The clinical instability group possessed significantly less IR ($P < 0.001$) and ER ($P = 0.03$) RoM (measured with a goniometer) than the control group. The findings from the glenohumeral laxity exam and symptom surveys are presented in Tables 2 and 3, respectively.

## Glenohumeral joint kinematics

The average amount of glenohumeral joint elevation was 97.16° for the clinical instability group and 109.21° for the control group. There was not a statistically significant interaction between the group and elevation angle for the humeral position on the anterior/posterior ($df = 4, 140$, $F = 0.48$, $P = 0.75$) or superior/inferior axes ($df = 4, 140$, $F = 0.45$, $P = 0.77$). Regarding the anteroposterior direction, a main effect of group was observed, with a statistically significant anterior shift detected for the clinical instability group ($df = 1, 140$, $F = 10.28$, $P < 0.001$, mean difference: 0.8 mm, $d = 0.57$). There was no significant difference in average humeral superior/inferior position between groups ($df = 1, 150$, $F = 0.60$, $P = 0.44$, mean difference: 0.22 mm, $d = 0.12$). A graphical representation of these results is shown in Fig 2.

There were no statistically significant interaction effects between group and phase of elevation for humeral positional dispersion ($df = 1, 56$, $F = 0.10$, $P = 0.76$) or CPL ($df = 1, 56$, $F = 0.02$, $P = 0.90$). There were no significant differences

**Table 1. Demographic and Clinical Characteristics.**

| Variable | Group | | P-value |
|---|---|---|---|
| | Control (n = 10) | Clinical Instability (n = 20) | |
| | Count/ Total (%) | Count/ Total (%) | |
| **Sex** | | | |
| **Female** | 6/ 10 (60%) | 13/ 20 (65%) | – |
| **Male** | 4/ 10 (40%) | 7/ 20 (35%) | – |
| **Side Examined** | | | |
| **Right** | 3/ 10 (30%) | 13/ 20 (65%) | – |
| **Left** | 7/ 10 (70%) | 7/ 20 (35%) | – |
| | Mean (range) | Mean (range) | |
| **Age (Years)** | 24.70 (20.00 - 43.00) | 24.85 (18.00 - 60.00) | 0.70 |
| **Height (m)** | 1.73 (1.57 - 1.91) | 1.76 (1.52 - 1.93) | 0.46 |
| **Weight (Kg)** | 71.35 (54.43 - 102.06) | 72.19 (55.79 - 98.45) | 0.43 |
| **BMI (Kg/m²)** | 23.63 (20.60 - 32.28) | 23.31 (19.39 - 28.32) | 0.44 |
| | Mean (∓SD) | Mean (∓SD) | |
| **External Rotation (°)** | 115 (11) | 101 (10) | < 0.01 |
| **Internal Rotation (°)** | 72 (11) | 65 (10) | 0.03 |

Abbreviations: SD: standard deviation; m: meter, Kg: kilogram, BMI: Body Mass Index.

between groups in positional dispersion ($df=1$, 56, $F=2.38$, $P=0.13$, mean difference: −5.93%, $d=0.37$) or CPL ($df=1$, 1, $F=1.30$, $P=0.26$, mean difference: −10.86%, $d=0.30$). There was no significant difference in CPL between groups ($df=1$, 1, $F=1.30$, $P=0.26$, mean difference: −10.86%, $d=0.30$). There was significantly greater humeral positional dispersion ($df=1$, 1, $F=23.04$, $P<0.01$, mean difference: 17.40%, $d=0.74$) across both groups during the first 50% of elevation and a trend towards significantly more CPL ($df=1$, 1, $F=3.88$, $P=0.05$, mean difference: 17.70%, $d=0.43$) during the second 50% of elevation. A graphical representation of these results is shown in Fig 3.

The raw data used to generate the results presented within our results section are available as a downloadable dataset in S1 Dataset.

## Discussion

This analysis aimed to compare the glenohumeral joint kinematics between individuals clinically classified with MDI and healthy, asymptomatic, matched controls during active arm elevation. We limited the clinical instability group to competitive swimmers to reduce inherent kinematic differences amongst our sample, who may otherwise have inconsistent exposures to overhead activities in recreation or occupation, and improve our ability to characterize the kinematic patterns suspected of MDI. Additionally, we used an advanced motion-tracking technique in dynamic biplane video radiography to overcome the limitations of surface-based motion-tracking systems and improve the accuracy of kinematic data. We also explored using novel kinematic variables to assess the differences in joint stability of participants with MDI compared to healthy controls. With this approach, we hypothesized that individuals classified with MDI would demonstrate significantly different kinematic patterns than asymptomatic controls.

The control group possessed significantly greater average ER (+14°) and IR (+7.0°) than the clinical instability group. While others suggest that competitive overhead athletes typically exhibit increased ER and decreased IR [49] in their sport-specific arm (e.g., pitching arm) compared to their other "control" arm, our results show that asymptomatic controls demonstrated greater ER and IR values than competitive swimmers classified with MDI. The increased range of motion in both ER and IR in controls likely reflects the absence of symptoms or apprehension that could restrict passive movement.

**Table 2. Findings from the Clinical Laxity Examination.**

| Grade | Group | |
|---|---|---|
| | **Healthy n=10** | **Clinical Instability n=20** |
| | Count/ Total (%) | Count/ Total (%) |
| **Composite Laxity Score** | | |
| <1.6 | 10/10 (100%) | 0/10 (0%) |
| >1.6 | 0/ 10 (0%) | 20/ 20 (100%) |
| **Anterior Drawer Test** | | |
| 0 | 1/ 10 (10%) | 0/ 20 (0%) |
| 1 | 6/ 10 (60%) | 1/ 20 (5.0%) |
| 2 | 3/ 10 (30%) | 19/ 20 (95%) |
| 3 | 0/ 10 (0%) | 0/ 20 (0%) |
| **Posterior Drawer Test** | | |
| 0 | 9/ 10 (90%) | 2/ 20 (10%) |
| 1 | 1/ 10 (10%) | 17/ 20 (85%) |
| 2 | 0/ 10 (0%) | 1/ 20 (5.0%) |
| 3 | 0/ 10 (0%) | 0/ 20 (0%) |
| **Sulcus Sign** | | |
| 1 | 10/ 10 (100%) | 0/ 20 (0%) |
| 2 | 0/ 10 (0%) | 15/ 20 (75%) |
| 3 | 0/ 10 (0%) | 5/ 20 (25%) |
| **Apprehension Sign** | | |
| **Negative** | 10/ 10 (100%) | 10/ 20 (50%) |
| **Positive** | 0/ 10 (0%) | 10/ 20 (50%) |
| | Mean (∓SD) | Mean (∓SD) |
| **Beighton's Index** | 1.00 (0.00) | 3.32 (1.73) |

**Table 3. Symptom Survey Results at Time of Collection – Clinical Instability Group.**

| Variable | Mean (SD or range) | Number of participants |
|---|---|---|
| **Days experience pain** | 28 (range: 6–68) | 20 |
| **VAS (0–10)** | 4.7 (SD:∓1.8) | 20 |
| **WOSI (%)** | 33.6% (SD:∓22.4) | 20 |

SD: standard deviation; VAS: Visual Analog Scale (0–10 points: 0=no pain, 10=worst); WOSI: Western Ontario Shoulder Instability questionnaire (0–100%: 0=no symptoms, 100=extreme symptoms).

The presence of competitive swimmers in the control group supports this concept, as one of these individuals possessed 84° of IR, which surpassed the mean plus one standard deviation for the control group (10.59°), yet did not report any symptoms nor have an increased composite laxity score. However, all other RoM measurements from these two competitive swimmers were within one standard deviation of the control group mean.

While including competitive swimmers in the control group potentially reduces bias when comparing kinematics between the two groups, a post-hoc analysis indicated that one swimmer contributed outlying kinematic data compared to other controls. Specifically, this participant exhibited outlier data at 45°, 60°, 75°, and 90° of glenohumeral elevation, characterized by greater anterior (at 45° only) and superior (at all angles) humeral positioning compared to the rest of the control group. This data is illustrated as a figure in S1 Fig 4 and S2 Fig 5. Despite the presence of outlier data points, their

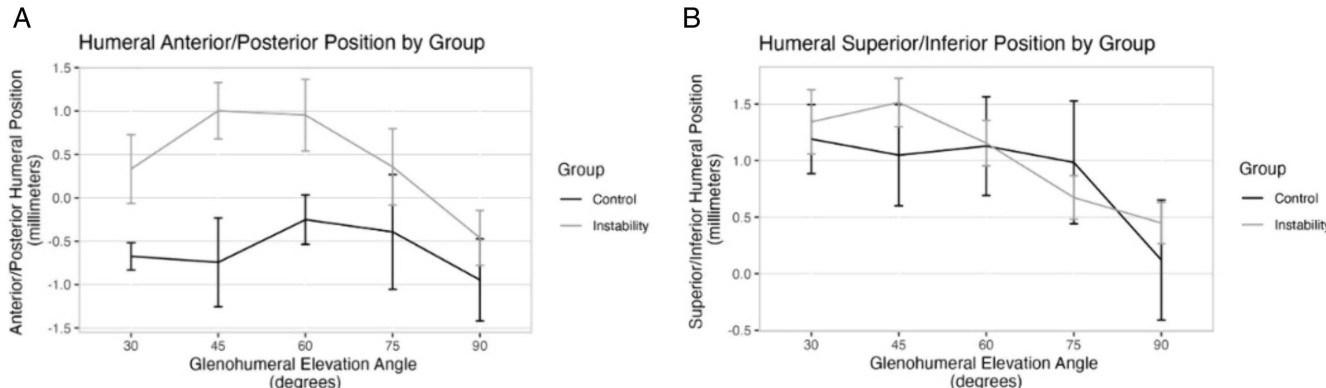

**Fig 2. Humeral Head Position by Glenohumeral Elevation Angle.** Caption: Data represents mean +/- standard error. The anterior and superior directions are positive on the Y axes, expressed relative to the glenoid center. 2-A: Humeral anterior/posterior position. 2-B: Humeral superior/inferior position.

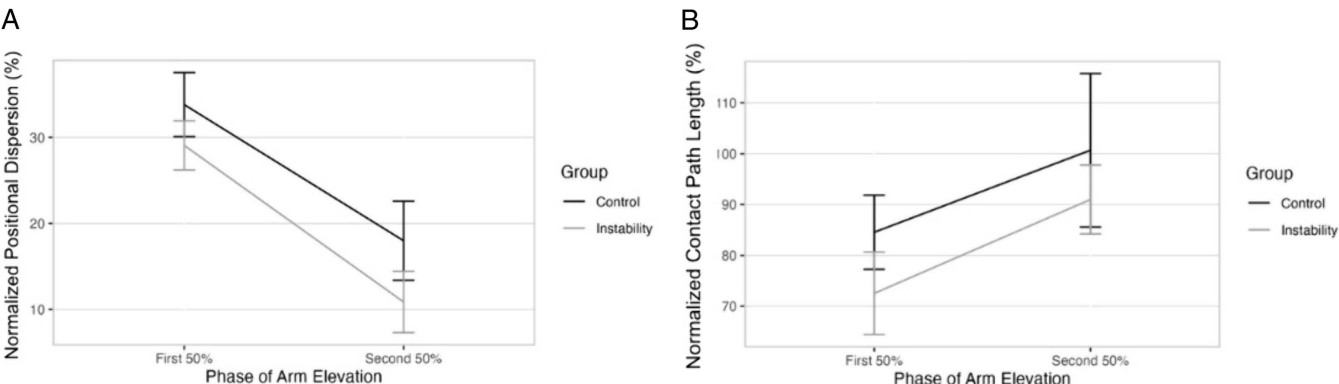

**Fig 3. Humeral Position Dispersion and Contact Path Length by Phase of Elevation.** Caption: Data represents mean +/- standard error. Humeral instantaneous helical axis (IHA) positional dispersion and contact path length (CPL) are normalized to glenoid height. 3-A: Humeral IHA positional dispersion. 3-B: CPL.

removal had no impact on the statistical outcomes. Overall, we believe that including competitive swimmers in the control group reduces the risk of bias in comparing kinematics between two sets of groups with different functional demands and increases the generalizability of our results.

We observed a significantly greater average anterior humeral position (+0.8 mm) across 30°-90° of glenohumeral elevation in the clinical instability group compared to the control group. This pattern is similar to results reported by Illyés & Kiss [9], who described a larger average anterior (0.07 mm more) and superior (0.13 mm more) distance between humeral and scapular centers of rotation in 15 individuals classified with MDI compared to 15 control individuals during SAB. However, the results reported by Illyés & Kiss [9] were not significant. In our analysis, the kinematic differences between groups is supported by the glenohumeral laxity assessment, which identified 95% (19/20) of swimmers with an increased anterior laxity grade of 2 or more. A more anteriorly positioned humeral head may pressure the highly innervated and vascularized labral periphery [50,51] and potentially impinge on anterior joint structures. Clinicians who treat overhead athletes may use these results to consider the impact of humeral position, assessed via clinical laxity examination, on anterior joint structures during repetitive overhead activity.

Despite detecting a significant kinematic difference between groups, the clinical implications of our findings are unclear. Previous research examining humeral position between symptomatic and asymptomatic individuals questioned if a 1.4 mm superior/inferior difference was clinically relevant relative to rotator cuff compression under the acromion [21]. In our analysis, we detected an average anterior/posterior difference of less than 1 mm across a 60° arc of motion. While statistically significant, the sub-millimeter difference falls below the observed measurement error (<0.9 mm [39]) for the 2D/3D shape-matching method. Ultimately, the kinematic difference between groups is minimal, and the suspected kinematic patterns of MDI are not yet completely determined. Future studies should consider using a similar approach to capture high-resolution dynamic kinematics, and also may consider measuring muscle activity during motion based on evidence that muscle activation ratios may differ between groups [9,52].

The lack of significant differences in humeral positions between groups highlights a limitation to our analysis of kinematics captured during unweighted SAB. The motion of unweighted SAB may not have been strenuous enough to induce the anticipated kinematic differences between groups. Analysis of glenohumeral joint kinematics captured during a more rigorous task, or potentially after a rigorous warm-up activity, may elicit the kinematic variability associated with MDI, especially considering competitive swimmers are used to prolonged resistance exercise during swim training activities. Further, analysis of data captured during the depression phase may offer valuable insight into the kinematics of MDI. Researchers may consider recording motion during more rigorous tasks, motions, or implementing a bout of shoulder activities designed to challenge the glenohumeral joint stabilization mechanism before motion capture.

Analysis of positional dispersion and CPL as measurements of joint stability presents challenges due to their novelty as kinematic variables. However, theoretical expectations provide context: in the most stable circumstance, the humeral axis of rotation would remain stationary (indicated by lower positional dispersion), and the humeral head would maintain the shortest path on the glenoid (indicated by lower CPL). In this context, higher positional dispersion and CPL values would be anticipated in shoulders classified with MDI, aligning with the clinical assumption that detrimental increases in humeral translations occur during arm activity.

Although this study did not reveal a significant interaction between groups and elevation phases for positional dispersion or CPL, notable variations emerged across the two elevation phases, regardless of group. Humeral positional dispersion was significantly greater during the first phase of elevation (+17.40%) compared to the second phase, and there was a trend toward significantly greater CPL during the second elevation phase compared to the first. In the first phase, < 45° glenohumeral elevation or ~68° humerothoracic elevation [46], there are lower joint loading and reaction forces – creating minimal challenges to joint congruency [53] thereby resulting in more fluctuation in the axis of rotation (more positional dispersion) and less concentrated contact between surfaces (less CPL). In contrast, as the arm moves through the second half of elevation (>70° humerothoracic elevation), suspected increased ligament tautness and joint loading provide a more effective constraint. In the context of our results, an average reduction of 17.40% in IHA positional dispersion, relative to the average glenoid height (31.74 mm), is equal to 5.52 mm less dispersion or movement from the pivot point, demonstrating a more stable axis. The added constraint on the humeral head during the second elevation phase would reduce osteokinematic dispersion. At the same time, the increased joint 'tightness' would promote more contact between joint surfaces and contribute to greater CPL as the humerus rolls efficiently on the glenoid.

We postulated that CPL was a valid kinematic descriptor of joint stability during SAB. For example, we assumed that a longer CPL indicates greater instability, reflecting increased humeral movement on the glenoid, based on research linking shorter CPL to enhanced joint stability [18,54]. However, this interpretation presumes that a longer CPL inherently signals a problem. Alternatively, a longer CPL might represent subtle, frequent humeral movements confined to a small area near the glenoid center. Our data may support this concept, with minimal difference in CPL observed between groups during SAB. This notion describes a paradox to the interpretation that a longer CPL indicates a more unstable joint. Future research aiming to utilize contact pattern variables to quantify joint stability may consider additional measures, such as the frequency and location of the contact center, in concert with CPL, to better differentiate stable from unstable joints.

We also compared humeral IHA positional dispersion between groups as a potential differentiator of stability, but did not observe significant differences. One possible explanation for these findings is the relatively small, single-plane motion occurring during SAB. For example, prior research has identified significant differences in helical parameters during large, multi-planar movements, such as cervical spine circumduction [55]. The lack of detectable differences in our study may indicate that the envelope of motion in SAB was insufficient to reveal subtle stability differences between groups. Additionally, the unweighted nature of the motion may not have adequately challenged either group to influence joint stability. To better assess stability, future kinematic analyses should explore helical positional dispersion during larger, less constrained movements.

The inclusion of competitive swimmers in our analysis is both a strength and a limitation. As a strength, this cohort represents individuals with consistent and high-volume participation in overhead activity, enhancing the relevance of our findings to populations engaged in repetitive shoulder loading. We intentionally selected competitive swimmers for the sample of individuals clinically classified with MDI to reduce within group inter-individual variability in shoulder kinematics, as prior research has shown that this population exhibits consistent training routines and movement patterns [15,16,50]. This approach allowed us to more effectively isolate the influence of clinical shoulder instability when comparing 3D glenohumeral joint kinematics during active scapular plane abduction (SAB) between swimmers with multidirectional instability (MDI) and matched controls without increased glenohumeral joint laxity and symptoms. However, limiting participants for the clinical instability group to competitive swimmers may have led to the enrollment of "copers." A coper is defined as an individual who either returns to, or continues to, participate in complete sports activities with an injury without undergoing surgical intervention [56]. The observation of glenohumeral joint kinematics captured from a sample of copers may have impacted our ability to elucidate the kinematic properties of MDI by capturing kinematics from a sample that can stabilize the humerus on the glenoid during activity, just as an asymptomatic control participant would. A potential way to address this in future research would be to enroll only competitive swimmers in both groups: competitive swimmers classified with MDI (increased glenohumeral joint laxity and symptoms) and competitive swimmers without increased glenohumeral joint laxity and no symptoms. In this scenario, copers may still be present but would be distributed across both groups, thereby minimizing the potential influence of coping status when comparing kinematics between suspected copers (i.e., competitive swimmers) and non-swimming controls.

A particular limitation of examining arthrokinematic variables is the surface accuracy of 3D reconstructed bone morphology. The 3D bone models for the clinical instability group were generated from CT images, while the 3D bone models for the control group were derived from MR images. This resulted from differing IRB protocols governing the two groups of participants' research participation. Specifically, competitive swimmers clinically classified with MDI were enrolled in research activities via an IRB protocol that contained CT imaging while participants for the control group were enrolled with an IRB that used MR imaging. CT data contains high-resolution images of cortical bone margins and limited soft-tissue information. In contrast, MR data includes both bone and soft-tissue information, but with sometimes indistinct contrast between cortical bone and soft-tissue structures [57]. Both modalities can reliably derive 3D participant-specific bone models for 2D/3D shape-matching [39]. However, 3D bone models from MR data are susceptible to including more cartilage in fully rendered models than those derived from CT images [57, 58]. A limitation of this analysis is comparing arthrokinematic data constructed from CT and MRI images, as small but impactful differences in bone surface generation may affect minimum distance calculations. Nonetheless, the difference in CPL between groups demonstrated a small effect size ($d$: 0.30; $P$=0.26). Therefore, we do not believe the imaging modality contributed significantly to our findings. Nevertheless, future research should employ a single 3D imaging technique when feasible to develop 3D bone models to ensure consistency in structural properties between participants and groups.

## Conclusion

In conclusion, we employed a state-of-the-art approach to examine glenohumeral stability between shoulders classified with MDI and controls. We detected a statistically significant difference in one of four kinematic variables compared

between groups, finding that those with instability demonstrated greater anterior position when performing glenohumeral elevation. However, the difference between groups was minimal and below the resolution of the motion capture and quantification method. We did not detect statistical differences between groups in humeral superior/inferior position, humeral instantaneous helical axis positional dispersion, or humeral contact path on the glenoid. Further exploration is necessary to identify kinematic variables that account entirely for the nuances of joint instability, enabling the assessment of the impact of different types of movement and interventions.

## Supporting information

**S1 Dataset. Downloadable File Containing Raw Data Used In Analysis.** This Microsoft Excel workbook contains the raw data used to complete our analysis. Separate sheets in the workbook contain the data for specific analyses. The order of data presented in each sheet is as follows: (A) Demographic, range of motion, and laxity examination data for all participants is presented in the Demographics sheet. (B) Symptom data for the clinical instability group at the time of kinematic data collection is presented in the Symptom Survey sheet. (C) Humeral position data during arm elevation is presented in the Osteokinematics sheet. (D) Humeral IHA positional dispersion and contact path length, covering the first and second 50% ranges of elevation, is presented in the Arthrokinematics sheet. (E) A variable code book is presented in the Variable Code Book sheet.
(XLSX)

**S1 Fig 4. Outlier Data From Control Swimmers in Humeral Anterior/Posterior Position.** Caption: triangles and squares represent individual data from the two competitive swimmers included in the control group compared to the IQR (vertical black lines) of the control group and control group means, indicated by the "X".
(TIF)

**S2 Fig 5. Outlier Data From Control Swimmers in Humeral Superior/Inferior Position.** Caption: triangles and squares represent individual data from the two competitive swimmers included in the control group compared to the IQR (vertical black lines) of the control group and control group means, indicated by the "X".
(TIF)

## Acknowledgments

The authors would like to thank Dr. Rebekah Lawrence for her assistance in processing the data.

## Author contributions

**Conceptualization:** Oliver Silverson, Ward M. Glasoe, Paula M. Ludewig, Justin L Staker.

**Data curation:** Oliver Silverson, Justin L Staker.

**Formal analysis:** Oliver Silverson, Gaura Saini, Ward M. Glasoe, Justin L Staker.

**Funding acquisition:** Oliver Silverson, Gaura Saini, Ward M. Glasoe, Justin L Staker.

**Investigation:** Oliver Silverson, Gaura Saini, Ward M. Glasoe, Paula M. Ludewig, Justin L Staker.

**Methodology:** Oliver Silverson, Gaura Saini, Ward M. Glasoe, Paula M. Ludewig, Justin L Staker.

**Project administration:** Oliver Silverson, Justin L Staker.

**Resources:** Oliver Silverson, Paula M. Ludewig.

**Software:** Oliver Silverson, Paula M. Ludewig.

**Supervision:** Oliver Silverson, Ward M. Glasoe, Paula M. Ludewig, Justin L Staker.

**Validation:** Oliver Silverson.

**Visualization:** Oliver Silverson.

**Writing – original draft:** Oliver Silverson, Gaura Saini, Ward M. Glasoe, Paula M. Ludewig, Justin L Staker.

**Writing – review & editing:** Oliver Silverson, Gaura Saini, Ward M. Glasoe, Paula M. Ludewig, Justin L Staker.

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
