## [Decision Letter · Decision Letter 0]

26 Jun 2025

Dear Dr. Silverson,

Thank you for submitting your manuscript to PLOS ONE. After careful consideration, we feel that it has merit but does not fully meet PLOS ONE’s publication criteria as it currently stands. Therefore, we invite you to submit a revised version of the manuscript that addresses the points raised during the review process.

**Please consider making following changes along with the changes suggested by the reviewers before we can consider the manuscript further :**

There is a difference in results and conclusion; result shows a significant change, but the conclusion is not the same.Define 3DAdd p-value wherever applicablePlease be consistent with the formatting.In L-82, the study is inanimate; please correct.Please explain the experimental design in detail.Please explain and report the statistical findings. 

We look forward to receiving your revised manuscript.

Kind regards,

Prateek Srivastav, PhD

Academic Editor

PLOS ONE

Journal Requirements:

Reviewers' comments:

Reviewer's Responses to Questions

**Comments to the Author**

1. Is the manuscript technically sound, and do the data support the conclusions?

Reviewer #1: Yes

Reviewer #2: No

2. Has the statistical analysis been performed appropriately and rigorously?

Reviewer #1: Yes

Reviewer #2: I Don't Know

3. Have the authors made all data underlying the findings in their manuscript fully available?

Reviewer #1: Yes

Reviewer #2: No

4. Is the manuscript presented in an intelligible fashion and written in standard English?

Reviewer #1: Yes

Reviewer #2: Yes

Reviewer #1: General comments

Thank you for the opportunity in reviewing the manuscript, it is a well-designed and executed study highlighting novelty within the area of shoulder pain in athletic populations.

I have very minor comments, around the consistency of writing and some of the statistical analyses, but it is well written manuscript with very few flaws.

Specific comments

Abstract

L21 – Define 3D on first use

L31 – If you state significantly, add the p value to the brackets on line 32.

L32 – You need a space between 0.8 and mm, there is inconsistency in this though the manuscript.

Introduction

L57 - Define 3D on first use

L82 – A study is inanimate and cannot compare, please amend accordingly.

Methods

L88-89 – This sentence seems out of place, consider starting with explaining the experimental design, then this ethics information would be more appropriate.

L153 – Check the writing style here “(35) (ICC: > 0.91) (36).”, it just needs work. Consider, combining the references so that the ICC is before them both.

L250 – What was the interpretation of the Cohen’s d effect size? Additionally, reporting the 95% CI would also be useful (but not necessary).

Results

L256 – Does this create an issue as the majority of the participants are non-swimmers, i.e. non-swimmers vs. swimmers?

Table 1 – you need to define if the numbers in brackets are the range, could some be the standard deviation?

Discussion

L313 and 391 – I would like to see some more consideration of the inclusion of only 2 competitive swimmers in the control group, this is potentially a factor that needs observation. Within the results, could you present individual data points where the swimmers could be identified. As you have primarily discussed the p-value, but highlight differences in these 2 control group participants.

Reviewer #2: The results mentioned a significant difference, so why is the conclusion that there is no difference? Perhaps the author could be more detailed in the abstract. Perhaps the author could be more detailed in the abstract.

**Do you want your identity to be public for this peer review?** For information about this choice, including consent withdrawal, please see our Privacy Policy

Reviewer #1: No

Reviewer #2: No

---

## [Author Response · Author response to Decision Letter 1]

22 Jul 2025

We thank the editor and reviewers for taking the time to manage and review our submission. We appreciate the feedback. As a team of authors, we have made the necessary modifications and believe our re-submission is stronger. Please see our specific responses to each reviewer in our Response to Reviewers letter. Thank you for taking the time to review our submission.

---

## [Decision Letter · Decision Letter 1]

13 Sep 2025

Dear Dr. Silverson,

Thank you for submitting your manuscript to PLOS ONE. After careful consideration, we feel that it has merit but does not fully meet PLOS ONE’s publication criteria as it currently stands. Therefore, we invite you to submit a revised version of the manuscript that addresses the points raised during the review process.

**Kindly make the changes suggested by the reviewer. **

We look forward to receiving your revised manuscript.

Kind regards,

Prateek Srivastav

Academic Editor

PLOS ONE

Journal Requirements:

Reviewers' comments:

Reviewer's Responses to Questions

**Comments to the Author**

Reviewer #3: All comments have been addressed

Reviewer #4: (No Response)

2. Is the manuscript technically sound, and do the data support the conclusions?

Reviewer #3: Yes

Reviewer #4: Partly

3. Has the statistical analysis been performed appropriately and rigorously?

Reviewer #3: Yes

Reviewer #4: Yes

4. Have the authors made all data underlying the findings in their manuscript fully available?

Reviewer #3: Yes

Reviewer #4: Yes

5. Is the manuscript presented in an intelligible fashion and written in standard English?

Reviewer #3: Yes

Reviewer #4: Yes

Reviewer #3: (No Response)

Reviewer #4: Dear Authors,

Thank you for the opportunity to review your manuscript, “Comparison of glenohumeral joint kinematics between swimmers clinically classified with multidirectional instability and asymptomatic controls.” The authors make an important contribution to the effort of identifying novel and more objective ways to classify multidirectional instability (MDI) in overhead athletes compared to asymptomatic controls. The paper is well organized, clearly written, and provides helpful considerations that will help pave the way for future studies using kinematic variables to detect MDI. Please see attached comments.

**Do you want your identity to be public for this peer review?** For information about this choice, including consent withdrawal, please see our Privacy Policy

Reviewer #3: **Yes: ** Dr. Suresh Sukumar

Reviewer #4: No

---

## [Author Response · Author response to Decision Letter 2]

1 Oct 2025

Dear Dr. Srivastav,

Thank you again for taking the time to review and manage our manuscript submission to PLOS ONE. As a team of authors, we have revised our manuscript in response to the comments from reviewer #4. We believe that our revised manuscript is stronger and clearer after making the suggested modifications, where necessary.

Below you will find our responses to each reviewer comment. The updated line numbers referenced in our responses below relate to the un-tracked version of our newly revised submission. Once again, thank you for taking the time to review our manuscript and manage our submission.

Best,

Oliver Silverson, PhD

Submitting author

Journal Requirements:

Journal Requirement

Please review your reference list to ensure that it is complete and correct. If you have cited papers that have been retracted, please include the rationale for doing so in the manuscript text, or remove these references and replace them with relevant current references. Any changes to the reference list should be mentioned in the rebuttal letter that accompanies your revised

manuscript. If you need to cite a retracted article, indicate the article’s retracted status in the References list and also include a citation and full reference for the retraction notice.

Response

Thank you for taking the time to review our manuscript and reviewer comments. After responding to the reviewer comments below, we did not adjust our reference list, but rather added context to our citation where the reviewer had commented.

We have reviewed our reference list and found no retracted citations.

Reviewer #4:

A general response from the authors (bold) to the general comment from reviewer #4: “Thank you for the opportunity to review your manuscript, “Comparison of glenohumeral joint kinematics between swimmers clinically classified with multidirectional instability and asymptomatic controls.” The authors make an important contribution to the effort of identifying novel and more objective ways to classify multidirectional instability (MDI) in overhead athletes compared to asymptomatic controls. The paper is well organized, clearly written, and provides helpful considerations that will help pave the way for future studies using kinematic variables to detect MDI. Please see my review comments below.”

The authors thank the reviewer for their thorough review of our submission and valuable feedback. We acknowledge your comments, critiques, and suggestions. We have addressed the specific comments in our revised submission and believe that our manuscript is improved. Our line-by-line responses to the specific comments from reviewer #4 are included below (please note that line numbers in our responses pertain to the untracked manuscript):

Reviewer comment

Recruitment timelines (Lines 104–106 & Lines 118–119)

Can you specify timelines for recruitment of the clinical instability group as part of the larger study compared to when controls were recruited?

(lines 118–119) state: “Participants for the clinical instability group were enrolled in a larger, ongoing research program investigating shoulder biomechanics and instability in competitive swimmers.” This does not seem to match the statement in lines 104– 106: “Participants were recruited through outreach to local swimming communities using digital and physical announcements, including targeted email campaigns, community flyers, and social media communications. Recruitment started on 01 September 2021 and ended on 28 February 2023.”

Response

We thank the reviewer for highlighting this description. Both groups (MDI group and control group) were enrolled and participated in all study activities related to the presented analysis between 01 September 2021 and 28 February 2023. A statement has been added to lines 97-98 to make this more clear.

Participants in the MDI group participated in subsequent research activities not relevant to the current analysis (e.g., treatment intervention, follow-up testing, etc.). The data captured from the MDI group presented in our submission was collected between 01 September 2021 and 28 February 2023. We added a statement to lines 113-115 to make this clear to the reader.

Reviewer comment

Power Analysis (Line 111)

Recommend adding how power analysis was conducted, e.g., which program/method did you use? (e.g., G*Power, R “pwr” package, etc.).

Response

We have added a statement to lines 107-109 to properly state and cite the “pwr” package used in R to complete our power analysis.

For consistency, we also added proper citation to the “mice” package used for multiple imputation, described in lines 254-256.

Reviewer comment

Recruitment rationale (Line 139)

It is unclear why clinical tests were used to classify MDI and then compared with imaging outcomes to see if differences in kinematic variables were observed, rather than using another method of imaging as a more direct reference standard. This approach risks creating a “comparison of a comparison,” since prior studies have already contrasted clinical classification with dynamic imaging. Please clarify the rationale for this framework and how it advances beyond existing work.

Response

We thank the reviewer for raising this question. As described in our text, we used a cluster of clinical laxity tests to classify individuals with or without increased glenohumeral joint (GH) laxity [1]. The grouping of these clinical laxity tests as a method to evaluate GH laxity was developed by several expert shoulder researchers and clinicians. Published data [2] has further established the criterion-validity of this approach to accurately identify individuals with multidirectional instability (increased GH laxity + symptoms) compared to controls when compared to a reference standard of dynamic fluoroscopic imaging.

We recognize that the initial validation of this approach was conducted with a relatively small sample size. However, we believe the sample was adequate to distinguish clinically meaningful subgroups, which provided sufficient justification to explore the utility of the kinematic variables examined in this study. Accordingly, we implemented this approach in our study to categorize individuals with or without increased GH laxity. Additionally, it is not reasonable to continue radiographic imaging during testing and motion capture due to radiation exposure for the participant and the researcher.

We have modified lines 128-130 to provide more context for our approach based on previous research.

1. Staker JL, Lelwica AE, Ludewig PM, Braman JP. Three-dimensional kinematics of shoulder laxity examination and the relationship to clinical interpretation. International Biomechanics. 2017;4: 77–85. doi:10.1080/23335432.2017.1372217

2. Staker JL, Braman JP, Ludewig PM. Kinematics and Utility of Shoulder Joint Laxity Tests as Diagnostic Criteria in Multidirectional Instability. Brazilian Journal of Physical Therapy. 2021;25: 883–890. doi:10.1016/j.bjpt.2021.10.001

Reviewer comment

Control group description (Line 164)

Here you state: “Competitive swimmers without a clinical classification of MDI and no shoulder pain were allowed to participate in the control group.”

This language is slightly confusing, as earlier you state that a clinical classification of MDI is present when the participant has MDI + pain, which makes it seem like a participant could be classified as having clinical MDI even if there is no shoulder pain present?

Response

We thank the reviewer for the ability to explain in further detail. The statement “Competitive swimmers without a clinical classification of MDI and no shoulder pain were allowed to participate in the control group” was originally written to describe the situation in which a swimmer, with no increased GH laxity or shoulder pain (i.e., a stable and non-painful shoulder), was allowed to participate in the control group. To make this statement more clear, we have modified the sentence to state: “Competitive swimmers without increased glenohumeral joint laxity and no shoulder pain were allowed to participate in the control group” (Line 150-152).

Related to the second point mentioned by the reviewer, we want to make it clear to the reviewer that only participants without increased glenohumeral joint laxity and no pain (i.e., only stable, asymptomatic, shoulders), were enrolled in the control group.

Reviewer comment

Rationale for IR/ER measurements (Lines 173–179)

Can you provide a further explanation of why IR/ER was included as a clinical characteristic? More specifically, can you expand on what you mean by: “These goniometric measures were used for clinically relevant group descriptions”?

Response

The goniometric measurements of IR and ER RoM were collected on all participants and presented in our manuscript as descriptive variables to define the movement characteristics of the two study groups. RoM measurements are typically presented in similar analyses and define the physical characteristics of unique samples, such as overhead athletes, and can often be valuable for group comparisons.

We have updated lines 163-164 to state this for the reader.

Reviewer comment

MR vs. CT reconstruction (Lines 200–202)

Can you provide a further explanation of why different methods of reconstruction were used for clinical instability (CT) and control groups (MR)?

Response

The 3D bone models for the clinical instability group were generated from CT images, while the 3D bone models for the control group were derived from MR images. The reason for the two different imaging modalities for each group was the result of the two IRB protocols governing this research. Specifically, participants enrolled in the clinical instability group were enrolled in a larger research effort beyond the scope of this analysis with an IRB that used CT imaging, while participants enrolled in the control group were enrolled via an IRB that had MR imaging only. Language has been added to lines 189-192 in the Methods section and again in lines 438-441 in the Discussion to make this clear to the reader.

Reviewer comment

Explain why active unweighted scapular plane abduction (SAB) was chosen (Methods)

I may have missed this, but it wasn’t clear to me why you chose SAB as the evaluated movement.

Can you please provide further justification of this movement in relation to your assessment procedure?

Response

The motion of SAB was selected since it is a commonly examined motion and reported on widely in the literature (See works: Spanhove, 2021; Generoso, 2025; Timmons, 2012). By capturing in vivo kinematics during a common motion, such as SAB, we can consider our findings in relation to other research. Furthermore, given that this study introduced novel descriptors of GH kinematics, this approach allowed us to focus on evaluating the new variables without simultaneously introducing a novel movement task.

We have added language to lines 170-172 to make our selection of SAB clearer to the reader.

Reviewer comment

Control group participants (Lines 294–295)

Lines 294–295 state: “Two participants in the control group were competitive swimmers without MDI.”

This statement seems to contradict the earlier inclusion/exclusion criteria for the control group. Does this mean 8/10 had asymptomatic MDI or were not competitive swimmers?

Response

We apologize for this confusion. We want to make it clear that participants in the control group were determined not to have increased GH laxity and no pain (as described in revised lines 150-152).

Participants in the control group could be swimmers, as was the case for the 2 individuals described in lines 294-295, however, these swimmers did not have increased laxity and had no pain (i.e., they had stable, asymptomatic shoulders). We have revised line 270-271 to make this clear to the reader.

Reviewer comment

Demographics Table (Line 300)

Can you provide a further explanation of why different methods of reconstruction were used for clinical instability (CT) and control groups (MR)?

Response

We believe that we have provided justification for who different imaging modalities were used on each group in response to reviewer comment #7.

Reviewer comment

Table 1. BMI MDI Group (Line 300)

Possible typo: upper range reported for BMI of 8.32 for clinical instability group.

Response

Thank you very much for catching this typo. The correct value is updated to “28.32”.

Reviewer comment

Table 2. Apprehension Sign Control (Line 302)

Check 0/10 (0%) negative value in table. Would expect most controls to have a negative apprehension sign.

Response

Thank you for pointing out this error. You are correct that 10/10 (100%) of the controls had a negative Apprehension sign. We have corrected this value in Table 2.

Reviewer comment

Statistical vs. Clinical Significance (Lines 311–313)

You report: “a main effect of group was observed, with the clinical instability group being significantly positioned anteriorly.”

As phrased, “significantly positioned” risks being misinterpreted as clinically significant, when your MCID was set at 1.0 mm and the observed difference (0.8 mm) was below both this threshold and the measurement error (<0.9 mm).

Recommend rewording to emphasize statistical significance only (e.g., “a statistically significant anterior shift was detected”).

Response

We thank the reviewer for their suggestion. We agree that alternate language could be used to describe these results. We have adjusted lines 288-290 accordingly.

Reviewer comment

Discussion summary (Lines 321–322 vs. 409–410)

Line 409 states: “During the first phase of elevation, humeral positional dispersion was significantly greater than in the second half (+17.40%)...”

This seems to contradict what is written in lines 321–322: “During the first 50% of motion, there was significantly less humeral positional dispersion (df = 1, 1, F = 23.04, P < 0.01).”

Can you please clarify this difference?

Response

We thank the reviewer for catching this oversight. To summarize our findings briefly: Across both groups, there was significantly greater humeral positional dispersion during the first 50% of elevation and a trend towards significantly more CPL during the second 50%.

In revisions, we have revised lines 298-301 in the Results and lines 382-384 in our discussion to correctly describe these results.

Reviewer comment

Comparison to Overhead Athlete Literature (Lines 346–351)

You write: “While others suggest that competitive overhead athletes typically exhibit increased ER and decreased IR [49], our results show that asymptomatic controls demonstrated greater ER and IR values.”

The cited reference (Wilk et al.) compares throwing vs. non-throwing arms, not pathological vs. healthy controls.

Consider clarifying this or citing studies that more directly compare symptomatic and asymptomatic athletes.

Response

We agree with the reviewer that the cited research could be described more clearly. We have modified lines 327-329 to make it clear that the cited work by Wilk et al pertains to throwing versus non-throwing arms and that our results compared asymptomatic controls to swimmers classified with MDI.

Reviewer comment

Discussion summary (Lines 426–430)

Although it is interesting that no significant changes between CPL were found between clinical MDI and control groups, I am not sure you can make the conclusion stated in lines 428–430: that the “data would support this concept with minimal difference in CPL between shoulders determined to be ‘unstable’ with clinical laxity testing compared to ‘stable’ control shoulders.”

As kinematic variables were tested in the present study using active shoulder abduction whereas clinical laxity testing was performed using manual maneuvers by an assessor passively loading the shoulder, which are entirely different, can you please clarify this difference?

Response

We agree that it could be more clear that we are discussing CPL during

---

## [Editor Report · Decision Letter 2]

6 Oct 2025

Comparison of glenohumeral joint kinematics between swimmers clinically classified with multidirectional instability and asymptomatic controls

PONE-D-25-28787R2

Dear Dr. Silverson,

We’re pleased to inform you that your manuscript has been judged scientifically suitable for publication and will be formally accepted for publication once it meets all outstanding technical requirements.

Kind regards,

Prateek Srivastav

Academic Editor

PLOS ONE
---

## [Editor Report · Acceptance letter]

PONE-D-25-28787R2

PLOS ONE

Dear Dr. Silverson,

I'm pleased to inform you that your manuscript has been deemed suitable for publication in PLOS ONE. Congratulations! Your manuscript is now being handed over to our production team.

Kind regards,

on behalf of

Dr. Prateek Srivastav

Academic Editor

PLOS ONE